# Ectopic expression of *WRINKLED1* in rice improves lipid biosynthesis but retards plant growth and development

Kaiqi Liu[1,2,3], Yuehui Tang[1,4], Yongyan Tang[1], Meiru Li[1,2], Guojiang Wu[1,2], Yaping Chen[1,2], Huawu Jiang[1,2]*

**1** Key Laboratory of Plant Resources Conservation and Sustainable Utilization, South China Botanical Garden, Chinese Academy of Sciences, Guangzhou, PR China, **2** Guangdong Provincial Key Laboratory of Applied Botany, South China Botanical Garden, Chinese Academy of Sciences, Guangzhou, PR China, **3** University of Chinese Academy of Sciences, Beijing, PR China, **4** College of Life Science and Agronomy, Zhoukou Normal University, Zhoukou, Henan Province, PR China

* hwjiang@scbg.ac.cn

**Data Availability Statement:** All relevant data are within the paper and its Supporting Information files.

## Abstract

WRINKLED1 (*WRI1*) is a transcription factor which is key to the regulation of seed oil biosynthesis in *Arabidopsis*. In the study, we identified two *WRI1* genes in rice, named *OsWRI1a* and *OsWRI1b*, which share over 98% nucleotide similarity and are expressed only at very low levels in leaves and endosperms. The subcellular localization of *Arabidopsis* protoplasts showed that *OsWRI1a* encoded a nuclear localized protein. Overexpression of *OsWRI1a* under the control of the CaMV 35S promoter severely retarded plant growth and development in rice. Expressing the *OsWRI1a* gene under the control of the P1 promoter of *Brittle2* (highly expressed in endosperm but low in leaves and roots) increased the oil content of both leaves and endosperms and upregulated the expression of several genes related to late glycolysis and fatty acid biosynthesis. However, the growth and development of the transgenic plants were also affected, with phenotypes including smaller plant size, later heading time, and fewer and lighter grains. The laminae (especially those of flag leaves) did not turn green and could not unroll normally. Thus, ectopic expression of *OsWRI1a* in rice enhances oil biosynthesis, but also leads to abnormal plant growth and development.

## Introduction

Angiosperm seeds accumulate storage protein, oil (mainly triacylglycerols, TAG) and carbohydrate (mainly starches) during seed filling and degrade these compounds upon germination in order to support early seedling growth. The relative abundance of storage reserves varies among seeds of different species. Many oilseeds produce 50–70% oil, some legumes contain 40% protein, whereas the seeds of most cereals have starch as 70–85% of the dry weight [1]. The accumulation of storage components requires the coordinated expression of many genes that encode the enzymes of the corresponding pathways [2]. Several transcription factors have been reported to play roles in regulating these pathways. One of the most important of these genes is *WRINKLED1* (*WRI1*) [3].

**Funding:** The research was supported by grants from the National Basic Research Program of China (973 Program) (2010CB126603), the Natural Science Foundation of Henan Province (202300410520), Key Scientific Research Projects of the Higher Education Institutions of Henan province (21A180028). The funders had no role in study design, data collection and analysis, decision to publish, or preparation of the manuscript.

**Competing interests:** The authors have declared that no competing interests exist.

The *WRI1* gene products belong to the AINTEGUMENTA (ANT) subfamily in AP2/ERF transcription factor family, members of which contain two plant-specific (AP2/EREB) DNA-binding domains [4]. The function of the *WRI1* gene in plants is first reported in *Arabidopsis*. *Arabidopsis wri1* mutant seedlings have no obvious phenotype during vegetative development, but they produce wrinkled seeds and the ability of developing seeds to convert sucrose and glucose into precursors of fatty acid (FA) biosynthesis is impaired, subsequently causing an 80% reduction in seed oil content [3]. As a transcription factor, the WRI1 protein can target, and promote the expression of, many genes encoding enzymes involved in late glycolysis and FA biosynthetic pathways [5, 6]. In *Arabidopsis* [4, 7], *Brassica napus* [8, 9], *Nicotiana benthamiana* [10], and *Solanum tuberosum* [11], overexpression or ectopic expression of *WRI1* genes increases the oil content of seeds, leaves, or tubers.

In maize, overexpression of *ZmWri1a* increases the fatty acid content of the mature grain. The higher seed oil content in *ZmWri1a* overexpressing lines is not accompanied by any reduction in starch. The transgenic kernels do not show any macroscopically visible defects such as wrinkled or plump phenotypes, and there is no significant difference in kernel weight between transgenic and wild-type kernels [12, 13]. The ectopic expression of *BdWRI1* in the grass *Brachypodium distachyon* induces the transcription of genes predicted to be involved in glycolysis and FA biosynthesis, and TAG content is increased up to 32.5-fold in 8-week-old leaf blades. However, ectopic expression of *BdWRI1* also cause cell death in leaves [14].

Rice (*Oryza sativa* L.) has two genes, LOC_Os11g03540 and LOC_Os12g03290, which are close to the *Arabidopsis WRI1* gene on a phylogenetic tree [15]. The LOC_Os11g03540 gene (AJ575217) is found to be mainly expressed in embryos, however, the expression of this gene is hardly detected in leaves and endosperms [16]. Sun et al. [17] expresses the *CoWRI1* gene of Coconut (*Cocos nucifera* L.) using an endosperm-specific promoter EnP2 from rice changes the seeds oil content in transgenic rice. In order to explore whether the expression of *OsWRI1* could increase oil accumulation in non-oil storage organs such as endosperms and leaves of rice plants, we overexpressed the LOC_Os11g03540 gene under the control of the CaMV 35S promoter. We observed that the resulting transgenic plants were severely retarded with respect to growth and development. Expressing this gene under the control of the P1 promoter of *Brittle2* (small subunit of ADP–glucose pyrophosphorylase) [18] increased the oil content of leaves and endosperm but affected plant growth and development in rice.

## Materials and methods

### Plant materials

Using the *japonica* rice (*Oryza sativa* L.) cv. Zhonghua 11 (ZH11) as the wild type materials. After the seeds of wild-type and transgenic plants germinated, they were planted in a glass greenhouse under natural light in the South China Botanic Garden, China. Roots, leaves, embryos and endosperms from seeds of plants 10 days after flowering were sampled, and stored at −80°C until required for expression profiles analysis.

### Sequence retrieval and analysis

TAIR database (http://www.*Arabidopsis*.org/) and Phytozome database (http://www.phytozome.net/) were used to download AINTEGUMENTA (ANT) amino acid sequences of *Arabidopsis* and rice, respectively. However, the ANT sequences of maize and *Jatropha curcas* were derived from GenBank (http://www.ncbi.nlm.nih.gov/). ClustalW software was used to analyze the full-length ANT amino acid' bioinformatics characteristics. And then the NJ (neighbor-joining) method was used to construct a rootless phylogenetic tree based on the Mega software version 5 [19].

## Subcellular localization of OsWRI1a protein

Using rice leaves as a template, we cloned the *OsWRI1a* gene (without stop codon) by RT-PCR technology. Subsequently, we ligated the sequenced correct *OsWRI1a* gene into the pBWA(V) HS-GLosgfp vector to construct the OsWRI1a-GFP fusion protein vector. We further co-transformed the constructed OsWRI1a-GFP fusion expression vector and empty vector into *Arabidopsis* protoplast cells by a PEG-mediated method [20], and observed under a confocal electron microscope.

## Plasmid constructs and plant transformation

Using rice leaf cDNA as template, specific primers were designed (primers given in S1 File), and the coding sequence of full-length *OsWRI1* gene was cloned by RT-PCR. The cloned and purified DNA sequence was connected to the pMD18-T vector (Takara, http://www.takara.com.cn/). The qualified recombinant cloning vector pMD18-T -OsWRI1 plasmid was digested with *Bam*H I and *Sal* I. Then, the DNA fragment cut by *Bam*H I and *Sal* I was connected to pCAMBIA1301 vector under the control of the CAMV 35S promoter (35S) by T$_4$ DNA ligase (*Eco*R I – 35S – *Sac* I, *Bam*H I – *OsWRI1a* – *Sal* I, *Pst* I – *ocs* – *Hind* III).

To construct an expression vector containing *OsWRI1a* under the control of the promoter of *Brittle2* (*Bt2*, endosperm ADP–glucose pyrophosphorylase small subunit; LOC_Os08g25734), the P1 promoter of the *Brittle2* gene was amplified by PCR from rice genomic DNA extracted from leaves (primers listed in S1 File) [18]. The correctly sequenced PCR product was connected to the pMD18-T vector, and the pMD18-T-OsWRI1 plasmid was digested by *Hinc* II and *Bam*H I. Then we linked the target gene to the pCAMBIA1301 vector through T$_4$ DNA ligase (containing *OsWRI1a* and ocs) at the *Sma* I/*Bam*H I site (*Sma* I/*Hinc* II – *Bt2*P1 – *Bam*H I – *OsWRI1a* – *Sa*l I, *Pst* I – *ocs* – *Hind* III). Next, we transformed the constructed *OsWRI1* plant expression vector into *Agrobacterium* (strain EHA105) by the freeze–thaw procedure. Finally, *Agrobacterium* containing *OsWRI1* plasmid was used to transform rice callus as described by Li and Li [21, 22], and obtained *OsWRI1* transgenic plants. In addition, the screening of homozygotes was based on a segregation ratio of about 3:1 observed in T2 plants. The segregation ratios of four transgenic plants, were confirmed by GUS staining, were 237:78, 197:64, 169:55 and 206:67, respectively. Then we randomly selected 30 T3 plants for GUS staining. We speculated that these plants were homozygous plants for all GUS-stained plants.

## RNA isolation and qRT-PCR

According to the instructions of the kit, plant total RNA was extracted by TRIzol reagent (Invitrogen, http://www.thermofisher.com). Then we took 2 μg total RNA, and reversed transcribe the RNA into cDNA through M-MLV reverse transcriptase (Promega, http://www.promega.com). An *OsRUB* (rice ubiquitin) gene was used as a reference. The LightCycler 480 (a Mini Option real-time PCR system) was used to perform qRT-PCR. Cycling conditions were as follows: 95˚C for 30 s, 95˚C for 5 s, 60˚C for 20 s, and 72˚C for 20 s. The reaction was carried out for 40 cycles. Three biological replicates were performed in this study, and average values were used to calculate the relative expression of genes [23, 24]. The primers were shown in S1 File.

## Chemical assays

Chlorophyll concentrations of samples were determined according to Arnon [25]. The chloroplast pigments were extracted from fully-expanded 2$^{nd}$ leaves with 80% acetone.

TAG and free FA analyses were performed as described by previously Yang et al. [14]. Total lipid samples were separated by TLC on silica plates (Si250PA; 20*20cm; Mallinckrodt Baker) developed with ether: ethyl ether:acetic acid (80:20:1, v/v). After development, TAG and free FA bands were sprayed with 0.01% (v/v) Primuline in 80% (v/v) acetone and visualized under UV light. TAG or free FA bands were isolated from the TLC plate. FA methyl esters were prepared and quantified as described previously [26].

For fatty acid composition analysis. To determine fatty acid composition, fatty acid methyl esters were prepared and quantitatively analyzed by gas chromatography according to Wu et al. [27]. An Agilent 7890A GC system apparatus with a flame ionization detector on an HP-88 column (30 m × 0.25 mm I.D., 0.20 μm film thickness) was used. Results are expressed as mol percentage [26]. Statistical analysis of the data was performed using Student's test.

# Results

## The *WRI1* genes of rice

*WRINKLED1* (*WRI1*) encodes a transcription factor of the AP2/ERF class. The protein has two plant-specific (AP2/ERF) DNA-binding domains, and in *Arabidopsis* it is involved in the control of storage compound biosynthesis [3, 4]. According to a phylogenetic tree based on AP2/ERF family proteins from selected plant species, seven AP2 proteins from rice fall into the ANTb clade which contains the AtWRI1 protein [15] (Fig 1). LOC_Os11g03540 and LOC_Os12g03290 proteins share the highest amino acid similarities with *Arabidopsis* WRI1 and maize WRI1 proteins, so we named them *OsWRI1a* and *OsWRI1b* respectively. The coding domain sequences of the two *OsWRI1* genes share very high (over 98%) nucleotide similarity. Their levels of expression could not be determined separately by RT-PCR. The expression levels measured by qRT-PCR in this study therefore represent the combined expression of the *OsWRI1a* and *OsWRI1b* genes, which were expressed mainly in roots and embryos (Fig 2). Of the two *OsWRI1* genes, we could only obtain cDNA clones for *OsWRI1a* by RT-PCR from total RNA of young seeds. Therefore, the *OsWRI1a* gene was selected for further functional verification.

## OsWRI1a localizes to the nucleus

In order to clarify the protein properties of OsWRI1a, we constructed an OsWRI1a-GFP fusion protein expression vector. Then we transferred the constructed vector and empty control vector into *Arabidopsis thaliana* protoplast cells by PEG-mediated method, and placed them under a fluorescence confocal microscope to observe the localization. The subcellular localization results showed that green fluorescence signal was detected in all cells with the empty control vector, while the fluorescence signal was only observed in the nucleus of the OsWRI1a-GFP fusion expression vector (Fig 3). The results show that the OsWRI1a protein is localized in the nucleus.

## Overexpression of *OsWRI1a* in rice seriously retarded plant growth and development

Ectopic expression of the *WRI1* gene in several plant species results in an increase in oil accumulation in leaves and other organs. To investigate whether ectopic expression of this gene exerted the same effect in rice, we overexpressed the *WRI1a* gene under the control of the CaMV 35S promoter. More than twenty transgenic lines were obtained. The transgenic seedlings exhibited multiple morphological defects, including dwarfism, chlorotic leaves, and failure to produce seeds. The upper parts of the laminae of most leaves were chlorotic and curly,

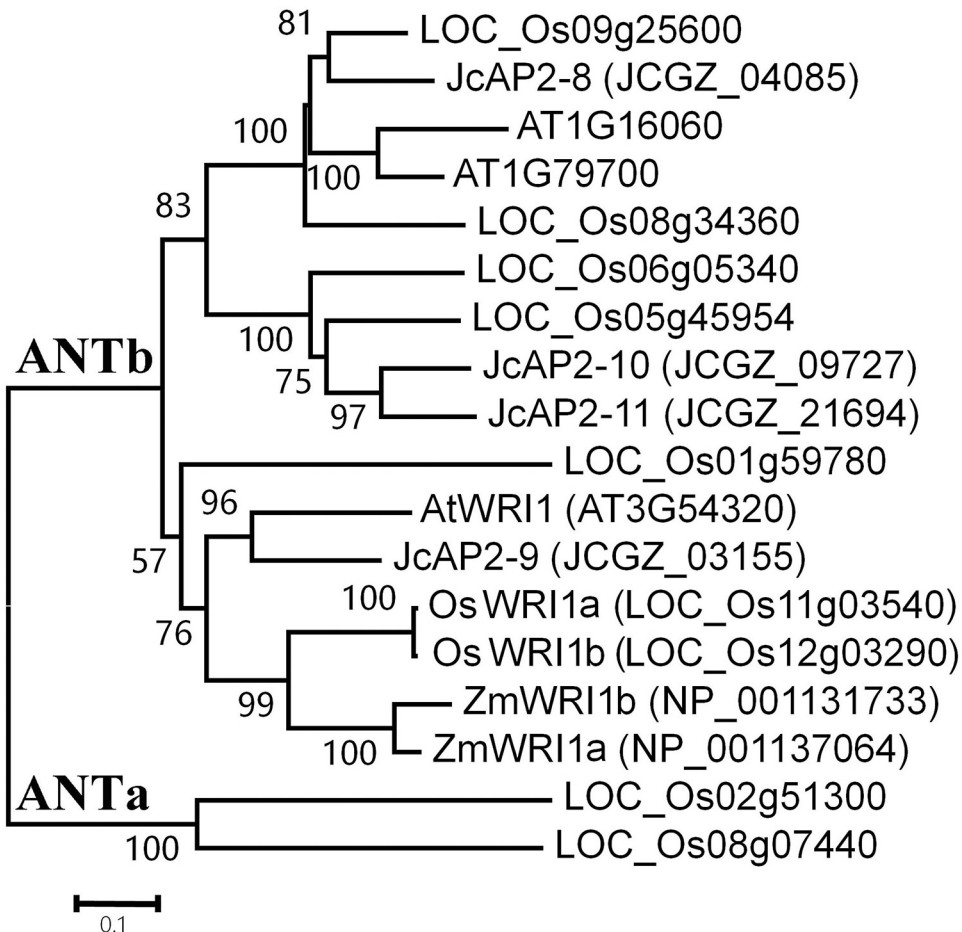

**Fig 1. Neighbor-joining unrooted tree.** Bootstrap values were calculated for 100 replicates, and values are indicated at the corresponding nodes. The database accession numbers of sequences are indicated in brackets.

and the chlorotic parts died quickly (S1 Fig). Because a high level of expression of the *OsWRI1a* gene seriously inhibited rice growth, and since expression of *ZmWRI1* under the control of the 19 KD ZEIN promoter (expressed at the later stage of endosperm development) do not lead to an increase in seed oil content in maize [12], we next expressed the *OsWRI1a* gene under the control of the P1 promoter of the *Brittle2* (*Bt2*) (LOC_Os08g25734) gene. The *Bt2* gene encodes two proteins, a cytosolic and a plastidial small subunit of ADP–glucose pyrophosphorylase. The GUS gene under the control of the *Bt2* P1 (P$_{Bt2P1}$) is highly expressed in endosperm but lower in leaves and roots of rice plants, whereas under the control of the P2 promoter, it is expressed strongly in roots and leaves but at a low level in endosperms [18]. Four independent P$_{Bt2P1}$::*OsWRI1a* transgenic lines (P1-6, -7, -8, and -10) were established for use in our experiments.

## Expression of *OsWRI1a* in rice under the control of the Bt2P1 promoter affected plant growth and development

The phenotypes of P$_{Bt2P1}$::*OsWRI1a* transgenic rice lines were observed in the T4 plants. It was found that P$_{Bt2P1}$::*OsWRI1a* transgenic rice plants were smaller and their heading date was about 7 days later than the wild type (ZH11) (Fig 4A–4C). The number of tillers in P$_{Bt2P1}$::

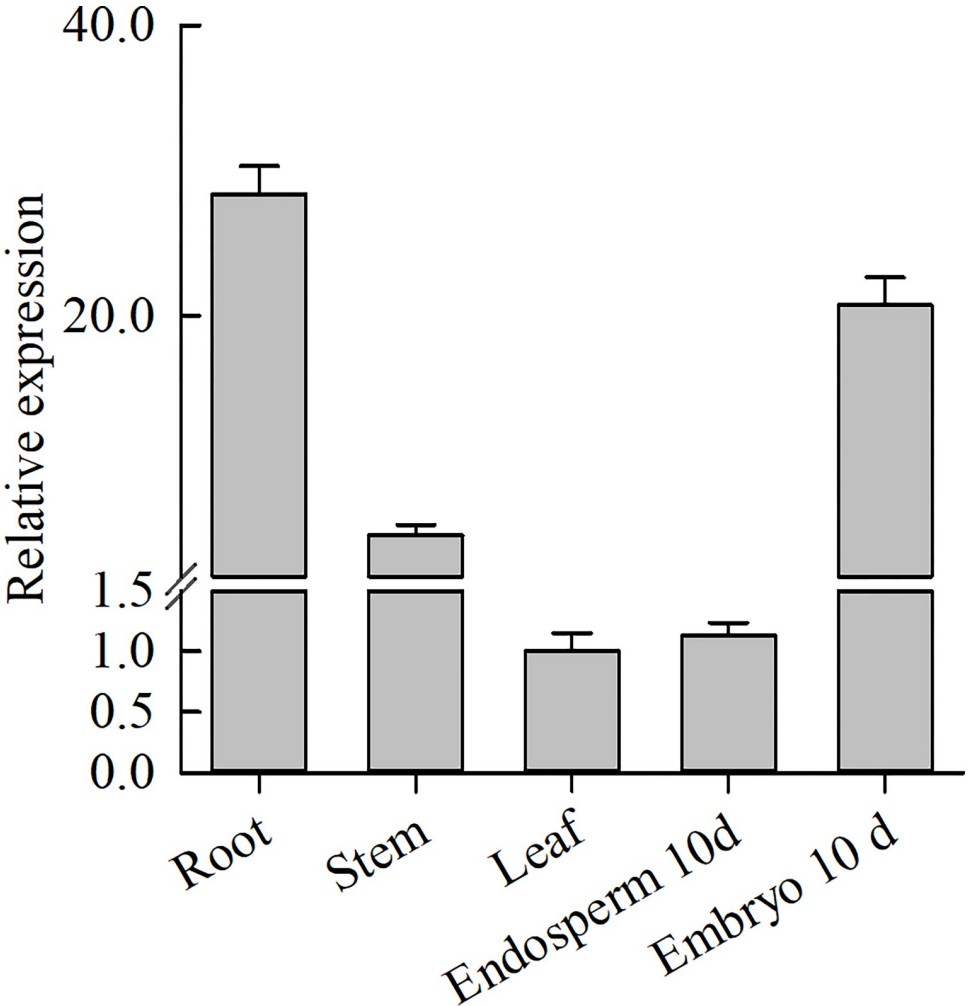

**Fig 2. Expression analysis of the *OsWRI1a/OsWRI1b* genes.** qRT-PCR applied to total RNA samples was used to measure *OsWRI1a/OsWRI1b* gene expression. Relative expression was normalized to that of the reference gene ubiquitin (internal control). The 10 d represent endosperms and embryos were sampled from seeds at 10 days after flowering. Bars show means ± SD of three biological replicates.

*OsWRI1a* plants (from 7.6±1.6 to 8.8±1.7) was less than that in the wild type (14.6±1.5) at the heading stage under present pot-culture conditions. The root lengths of P$_{Bt2P1}$::*OsWRI1a* seedlings were about 5% less than the wild type (S2 Fig). The P$_{Bt2P1}$::*OsWRI1a* plants had shorter internodes, shorter panicles and fewer grains than wild type plants (Fig 4E and 4F). The middle part of the lamina of all flag leaves and a small proportion of the second leaves did not turn green and failed to unroll properly during the period from emergence to death (Fig 5A–5D). Laminae of the second leaves were shorter, looked yellowish-green (Fig 5E) and contained less chlorophyll compared to the leaves of the same part of the wild-type plants (Fig 5F). Changes in levels of *OsWRI1a* transcripts in leaves were observed by qRT-PCR analysis (Fig 4D). The levels of *OsWRI1a* transcripts in the first leaf (flag leaf) was higher than that in the second and third leaves at the flag leaf stage of about 10 cm in length in P$_{Bt2P1}$::*OsWRI1a* transgenic plants (S3A Fig).

The crude lipid content of the leaves of P$_{Bt2P1}$::*OsWRI1a* plants was increased by 40% to 65% (S3B Fig), and the expression of several genes involved in later glycolysis and fatty acid

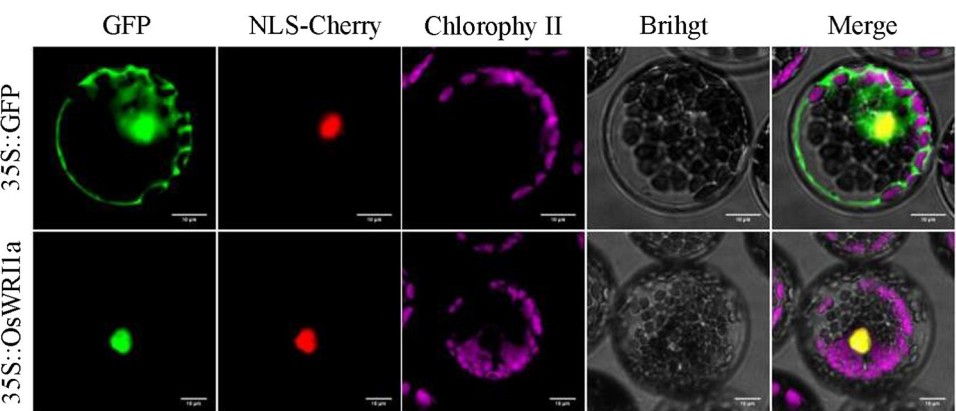

**Fig 3. Subcellular localization of OsWRI1a.** Transiently expressed the GFP (upper) and OsWRI1a-GFP fusion protein (lower) in *Arabidopsis* protoplasts which observed under a laser scanning confocal microscope. Bars = 10 μm.

biosynthetic pathway was upregulated (S3C Fig). To explore whether the levels of leaf TAG was also increased, we examined the content of TAG in the leaf blades of two $P_{Bt2P1}$::*OsWRI1a* lines. As a result, the content of TAG in $P_{Bt2P1}$::*OsWRI1a* leaf blades was increased by more

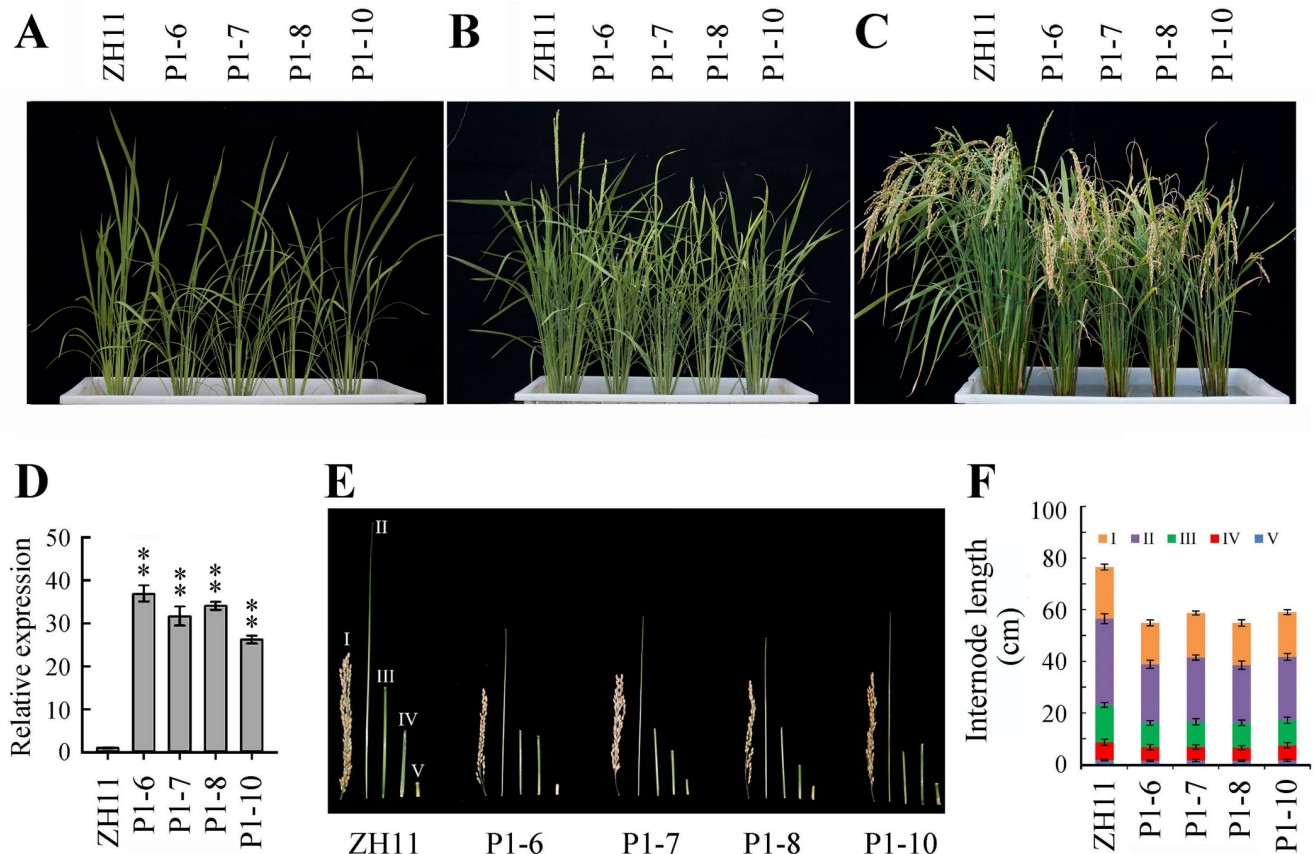

**Fig 4. The phenotypes of P$_{Bt2P1}$::*OsWRI1a* plants.** (A) Plants at the jointing-booting stage. (B) Plants at heading stage. (C) Plants at the mature stage. (D) Expression analysis of the *OsWRI1a* gene in leaves of plants at heading stage. The experiment included three biological replicates, each with two technical replicates. Values represent means of n = 6 ± SD (Duncan test: **, P < 0.01). (E) and (F) Lengths of internodes and panicles of the main stems. Values in (F) represent means of n = 30 ± SD. P1-6, -7, -8, and -10, P$_{Bt2P1}$::*OsWRI1a* transgenic lines.

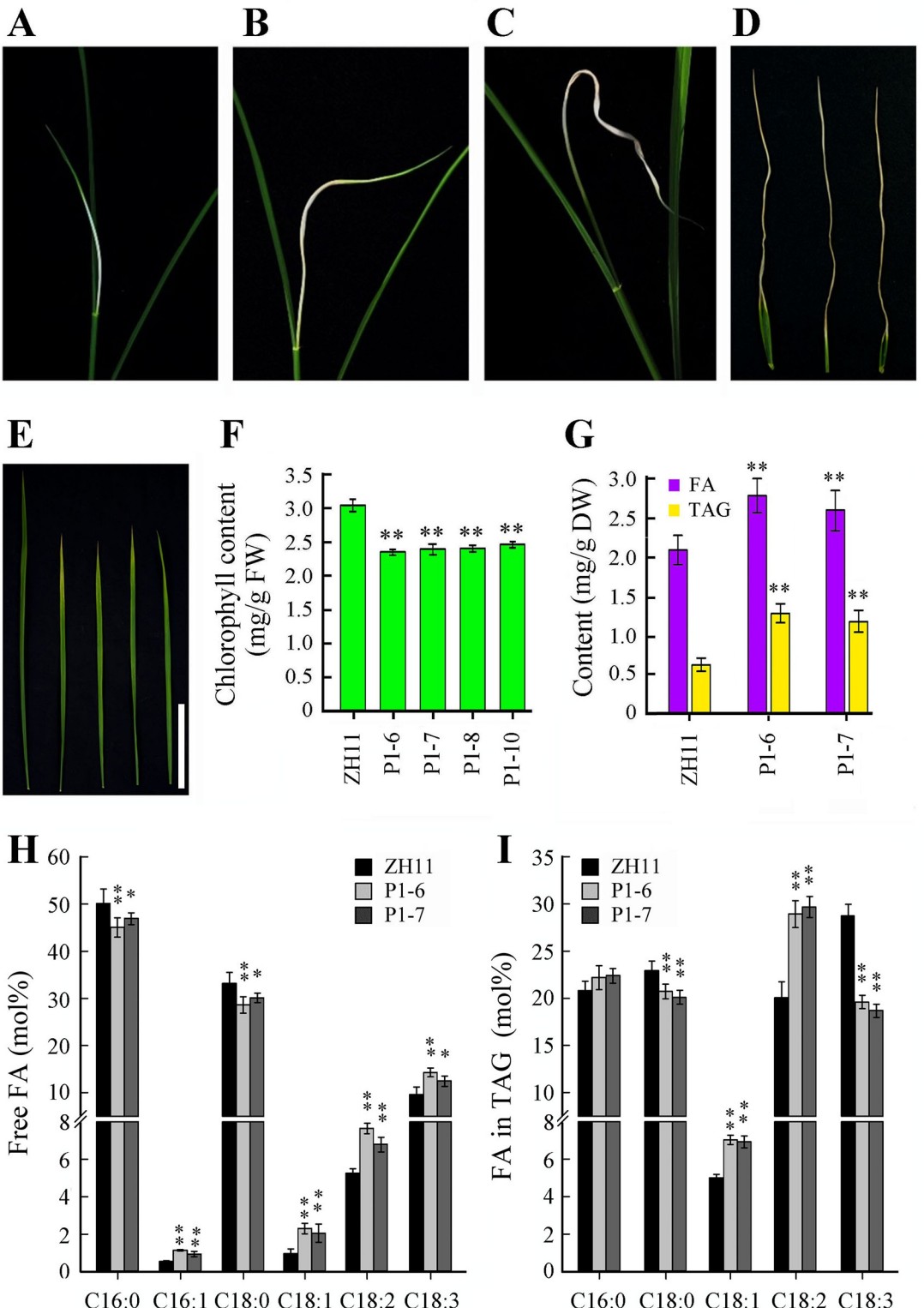

**Fig 5. The abnormal phenotypes of leaves in P$_{Bt2P1}$:: *OsWRI1a* plants.** (A) to (D) The developing course of abnormal lamina of flag leaves in P$_{Bt2P1}$:: *OsWRI1a* plants. P1-6, -7, -8, and -10, P$_{Sh2P1}$:: *OsWRI1a* transgenic lines. (E) The lamina of the second leaf of heading stage plants. Bar = 10 cm. (F) The chlorophyll content in the second leaves of heading stage plants. Values represent means of n = 6 ± SD (Duncan test: **, P < 0.01). (G) The FA and TAG content in leaves of heading stage plants. Values represent means of n = 3 ± SD (Duncan test: **, P < 0.01). DW = dry weight. (H) and (I) FA composition in free FA (H) and TAG (I) in

leaf blades of heading stage plants. Values represent means of n = 3 ± SD (Duncan test: **, P < 0.01). C16:0, palmitic acid; C16:1 palmitoleic acid; C18:0, stearic acid; C18:1, oleic acid; C18:2, linoleic acid; C18:3, linolenic acid. DW = dry weight. P1-6 and -7, P$_{Sh2}$P1::*OsWRI1a* transgenic lines.

than 40% (Fig 5G). After analyzing the fatty acid composition, we observed that the TAG of the P$_{Bt2}$P1::*OsWRI1a* lines contained more C18:1 and C18:2, but less C18:0 and C18:3 compared to wild-type lines (Fig 5I). Similar to the ectopic expression of *BdWRI1* in *B. distachyon* [14], the total free fatty acid content and total unsaturated fatty acid content was also increased in P$_{Bt2}$P1::*OsWRI1a* leaf blades (Fig 5G and 5H). These results indicate that the expression of *OsWRI1a* influences fatty acid biosynthesis in leaves and plant growth and development in rice.

### The P$_{Bt2P1}$::*OsWRI1a* lines had increased TAG content and altered fatty acid composition in endosperms

The 1000-seed weight of P$_{Bt2}$P1::*OsWRI1a* transgenic rice plants was approximately 15% lower than those of wild type (S4A Fig). The total crude lipid content of the endosperm of the mature seeds from P$_{Bt2}$P1::*OsWRI1a* lines was about 75% more than that of the wild type, while the net crude lipid content was no more than 1% (S4B Fig). We also analyzed the TAG content of endosperm and embryo of two P$_{Bt2}$P1::*OsWRI1a* lines. The TAG content in the endosperm of mature seeds from P$_{Bt2}$P1::*OsWRI1a* lines was over 30% more than that of the wild type (Fig 6A). The C16:0, C18:0, C18:1, and C18:2 fatty acids were the main components of TAG of rice endosperm. The TAG of endosperm from P$_{Bt2}$P1::*OsWRI1a* lines contained more C16:0, but less C18:1, compared to wild type (Fig 6B).

### Expression of *OsWRI1a* upregulated the expression of glycolysis and fatty acid biosynthesis related genes in rice endosperm

In maize and many other plants, overexpression of the *WRI1* gene upregulates the expression of many genes involved in glycolysis and in fatty acid biosynthetic pathways [6, 8, 12]. Therefore, we further examined the expression of several genes related to these pathways in the endosperm 5 and 10 days after flowering (DAF) by qRT-PCR. Compared to wild type, expression of 8 genes, *PK*, *PDH-E1 α*, *PDH-E1β*, *PDH-E2*, *KASIII*, *ENR1*, *ACP*, and *FatA*, was increased more than 2-fold at 5 DAF and/or 10 DAF (Fig 7).

## Discussion

The WRI1 protein is a key regulator of oil biosynthesis first discovered in plants [3, 28, 29]. Mutation in the *WRI1* gene causes a reduction in oil accumulation and wrinkled seeds. According to a phylogenetic analysis of AP2/ERF proteins, *Arabidopsis* has 3 AP2 proteins, while rice has 7 AP2 proteins that fall into the ANTb clade (Fig 1) [15]. As in the case of maize, the *WRI1* gene is duplicated in rice. The *OsWRI1* genes share very high nucleotide similarity, probably as a result of a recent segmental duplication between chromosomes 11 and 12 [30]. The *OsWRI1* genes were barely expressed in leaves and endosperms (Fig 2) [16], but their expression was high in embryos, suggesting that they may play roles in the biosynthesis of storage fatty acids in rice embryos.

Partial *Arabidopsis* transgenic lines (106 and 107) overexpressing *AtWRI1* show abnormal morphological phenotypes on agar plates. The seedlings do not turn green, lack expanded cotyledons, frequently do not open their apical hooks and often show elongated hypocotyls [4]. In the present study, we found that the upper parts of the laminae of most leaves in 35S::*OsWRI1a*

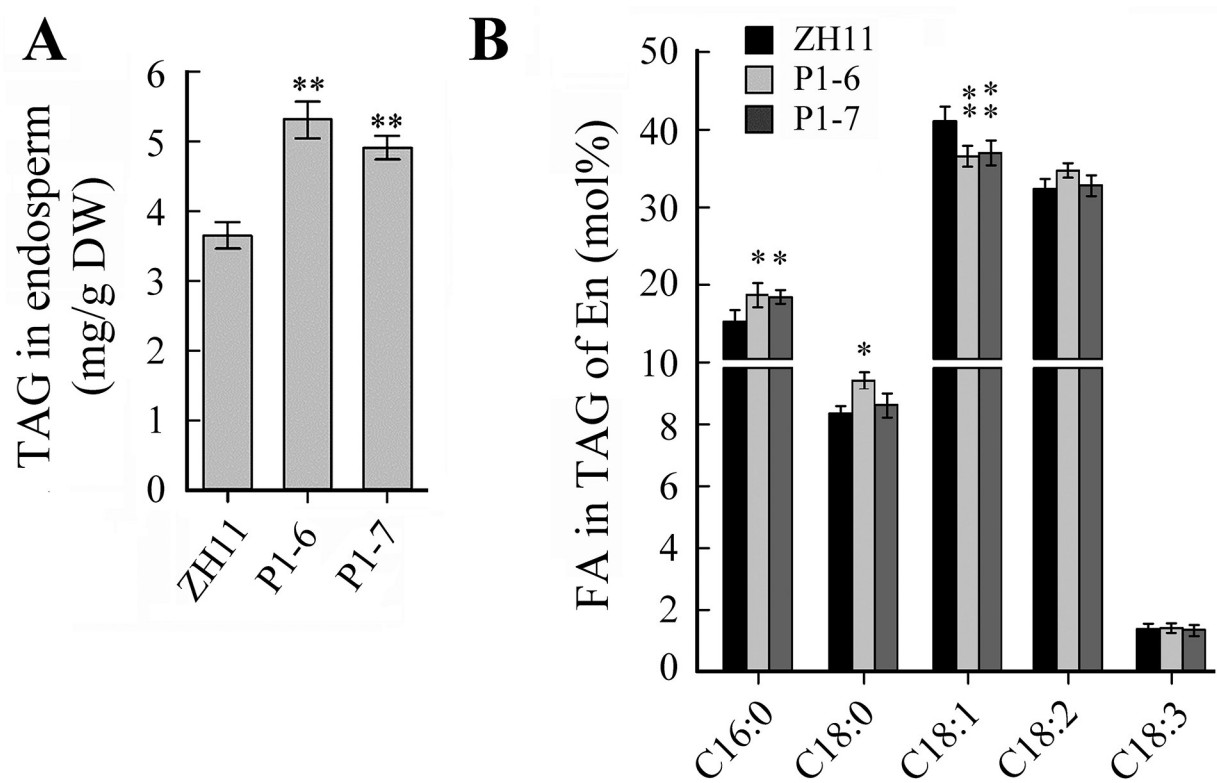

**Fig 6. Changes in TAG accumulation in endosperm and embryo.** (A) The TAG content in endosperm of mature seeds. (B) FA composition in TAG of endosperm. Values represent means of n = 3 ± SD (Duncan test: **, P < 0.01; *, P < 0.05). DW = dry weight. P1-6, and -7, P$_{Sh2P1}$::*OsWRI1a* transgenic lines. C16:0, palmitic acid; C18:0, stearic acid; C18:1, oleic acid; C18:2, linoleic acid; C18:3 arachidic acid.

expressing rice did not turn green and could not unroll. The growth and development of the transgenic seedlings were severely retarded (S1 Fig). In rice expressing P$_{Bt2P1}$::*OsWRI1a*, growth and development were less strongly retarded. Laminae of the flag leaves did not turn green and failed to unroll (Fig 5A–5D). These effects should base on the abundance of *OsWRI1* transcripts in different leaves of P$_{Bt2P1}$::*OsWRI1a* transgenic lines (S3A Fig). These results implied that a higher expression level of the *OsWRI1* gene could inhibit chlorophyll biosynthesis and photomorphogenesis in transgenic rice leaves, as was suggested in the case of partial *Arabidopsis* overexpressing *WRI1* lines under the control of the 35S promoter [4]. The development of rice leaves may be more sensitively to this inhibitory effect than *Arabidopsis* because it appears in all 35S::*OsWRI1a* expressing rice plants. Previous studies suggest that overexpression of *AtWRI1* in rice shows a significant increase in vegetative tissue fatty acids, with some seed reserve changes, but no phenotypic off types observed [31, 32]. We speculate that the different phenotypes of *AtWRI1*-overexpressing rice plants and *OsWRI1*-overexpressing rice plants may be attributed to the fact that genes in different species may play different functions, even though these genes are capable of increasing fatty acid content and are highly homologous. Similarly, *JcERF011* functions differently in *Arabidopsis* and rice [15]. In maize, overexpression of *ZmWRI1a* results in kernels that have elevated oil content, but do not show any visible defects with respect to seed yield and other agronomic traits [12, 13]. In *Brassica napus*, overexpression of *BnWRI1* accelerates flowering and enhances oil accumulation in both seeds and leaves without leading to any visible inhibition of growth [9]. Ectopic expression of *BdWRI1* in the grass *Brachypodium distachyon* induces the transcription of genes

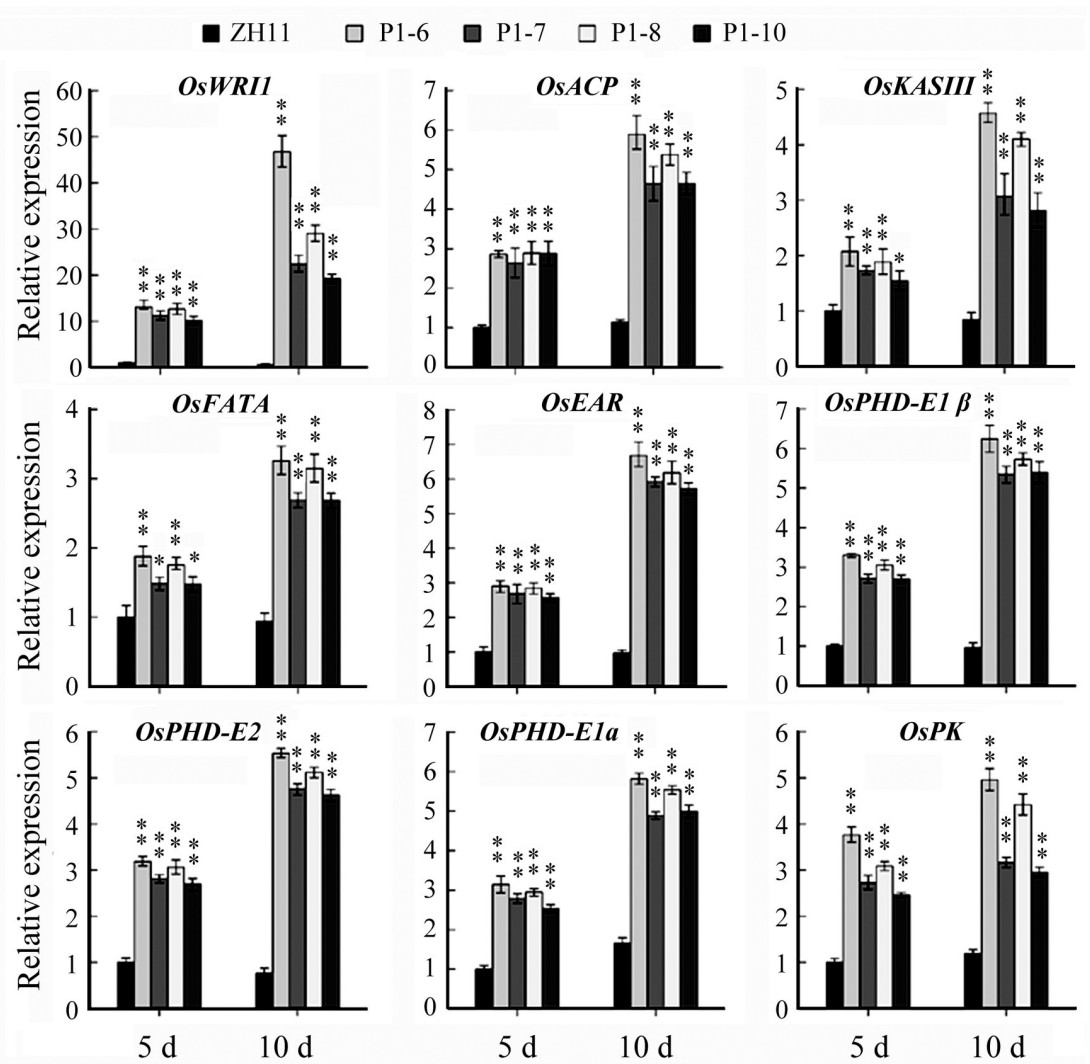

**Fig 7. Expression of genes predicted to encode proteins involved in the later stages of glycolysis and in fatty acid biosynthetic pathways in developing endosperms.** Total RNA was isolated from endosperms of the seeds of plants 5 and 10 days after flowering (5 d, 10 d). The experiment included three biological replicates, each with two technical replicates. Values represent means of n = 6 ± SD (Duncan test: **, P < 0.01; *, P < 0.05). P1-6, -7, -8, and -10, P$_{Bt2P1}$::OsWRI1a transgenic lines. The gene accession numbers are: OsACP, LOC_Os08g43580; OsENR1, LOC_Os08g23810; OsFatA, LOC_Os09g32760; OsKASIII, LOC_Os04g55060; OsPDH-E1α, LOC_Os04g02900; OsPDH-E1β, LOC_Os12g42230; OsPDH-E2, LOC_Os12g08170; OsPK, LOC_Os10g42100; OsRUB1, LOC_Os06g46770.

predicted to be involved in glycolysis and FA biosynthesis, and TAG content is increased in leaf blades. However, ectopic expression of *BdWRI1* also cause cell death in the leaf blades; this may be due to the high content of free fatty acids [14]. Taking these findings together, it appears that the specific phenotypes resulting from ectopic expression of the *WRI1* gene depend on plant species even within the same family.

As in other plant species, the *WRI1* gene has a function in FA biosynthesis in *Arabidopsis* and rice [8, 12, 33]. The expression of *OsWRI1a* under the control of the P1 promoter from the *Bt2* gene was sufficient to increase the lipid content of rice leaves and endosperms (Figs 5G and 6A). The underlying mechanism involves the transcriptional up-regulation of genes related to late glycolysis and fatty acid biosynthesis in leaves and endosperms (S3C Fig; Fig 7).

An unresolved question was why, although the oil content was increased, the net content in mature endosperms was no more than 1% (Fig 6A and S4B Fig). The same phenomenon was also observed in maize seeds as a result of overexpression of *ZmWRI1a* [12, 13]. The first possible explanation is the cytosolic localization of ADP–glucose pyrophosphorylase in rice and maize endosperms, which is responsible for partitioning large amounts of carbon into starch [34]. Another possibility is that the WRI1 protein regulates mainly the expression of the genes involved in glycolysis and FA biosynthesis, not those of the TAG assembly pathway. The third possibility is that, in oilseeds, the accumulation of oils in endosperms or cotyledons accompanies the degradation of starch and the assembly of TAGs [2, 26]. In contrast, TAGs seem to be rapidly turned over in leaves in *B. distachyon* plants overexpressing *BdWRI1* [14]. Thus, ectopically expressing solely the *OsWRI1* gene in rice, another grass species, would not be expected to result in the large-scale accumulation of TAG in leaves or endosperms.

## Conclusion

In this study, we cloned the *OsWRIa* gene and analyzed its function in rice. The *OsWRI1a* gene encoded a nuclear localization protein. *OsWRI1a*-overexpressing plants not only increased oil content in transgenic rice leaves and endosperm, but also resulted in abnormal growth and development of transgenic plants. This result adds to our understanding of the AP2 family gene *WRI1* in plant growth and development.

## Supporting information

**S1 File. Primers used in this study.**
(XLSX)

**S1 Fig. Phenotype of transgenic plants carrying 35S::*OsWRI1a* constructs.** Putative transformants were selected by detecting GUS reporter gene expression by means of a GUS staining assay. The (three-month-old) seedlings exhibited dwarfism, the upper parts of the laminae of most leaves were chlorotic and curly, and these leaves died more rapidly than those of wild-type plants.
(JPG)

**S2 Fig. Roots of 3-leaf stage in wild-type and transgenic plants carrying P$_{Bt2P1}$::*OsWRI1a* transgenic constructs.**
(JPG)

**S3 Fig. Lipid content and expression of genes predicted to encode proteins involved in the later stages of glycolysis and fatty acid biosynthetic pathways in leaves.** (A) The levels of *OsWRI1* transcripts in flag leaf (1$^{st}$ leaf), the second (2$^{nd}$ leaf) and the third (3$^{rd}$ leaf) leaves at the stage of the flag leaf was about 10 cm in length in two P$_{Bt2P1}$::*OsWRI1a* transgenic lines of P1-6 and P1-7. (B) The crude lipid content in leaves of heading stage plants. Values represent means of n = 6 ± SD (Duncan test: **, P < 0.01). DW = dry weight. (C) Expression of genes predicted to encode proteins involved in the later stages of glycolysis and fatty acid biosynthetic pathways in leaves. Total RNA was isolated from the second leaves of plants at heading time. The experiment included three biological replicates, each with two technical replicates. Values represent means of n = 6 ± SD (Duncan test: **, P < 0.01).
(JPG)

**S4 Fig. Changes in grain weight and lipid accumulation in endosperm.** (A) The 1000-seed weight. Seed weights were calculated by randomly selected seeds. The mature seeds were dried under 37˚C in an oven for three days. Values represent means of n = 3 ± SD (Duncan test: **,

P < 0.01). (B) The crude lipid content in endosperm of mature seeds. Values represent means of n = 6 ± SD (Duncan test: **, P < 0.01; *, P < 0.05). DW = dry weight.
(JPG)

**S1 Data.**
(XLSX)

## Acknowledgments

The authors appreciate the editor and reviewers for their helpful suggestions.

## Author Contributions

**Data curation:** Kaiqi Liu, Yuehui Tang, Yongyan Tang.

**Formal analysis:** Kaiqi Liu, Yongyan Tang, Meiru Li, Yaping Chen.

**Investigation:** Kaiqi Liu.

**Writing – original draft:** Kaiqi Liu.

**Writing – review & editing:** Guojiang Wu, Huawu Jiang.

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
