## [Decision Letter · Decision Letter 0]

7 Mar 2022

PONE-D-22-01261Ectopic expression of WRINKLED1 in rice improves lipid biosynthesis but retards plant growth and developmentPLOS ONE

Dear Dr. Jiang,

Thank you for submitting your manuscript to PLOS ONE. After careful consideration, we feel that it has merit but does not fully meet PLOS ONE’s publication criteria as it currently stands. Therefore, we invite you to submit a revised version of the manuscript that addresses the points raised during the review process.

Please make sure to address all concerns raised by the reviewers.

We look forward to receiving your revised manuscript.

Kind regards,

Tamar Juven-Gershon, Ph.D.

Academic Editor

PLOS ONE

Journal Requirements:

- https://academic.oup.com/jxb/article/56/412/623/580350?login=false

- https://academic.oup.com/plphys/article/156/2/674/6108701

- https://facultyopinions.com/prime/ext/725820437

The text that needs to be addressed involves the Introduction

In your revision ensure you cite all your sources (including your own works), and quote or rephrase any duplicated text outside the methods section. Further consideration is dependent on these concerns being addressed.

" The research was supported by grants from the National Basic Research Program of China (973 Program) (2010CB126603), the Natural Science Foundation of Henan Province (202300410520), Key Scientific Research Projects of the Higher Education Institutions of Henan province (21A180028)."

Reviewers' comments:

Reviewer's Responses to Questions

**Comments to the Author**

1. Is the manuscript technically sound, and do the data support the conclusions?

Reviewer #1: Partly

Reviewer #2: Partly

2. Has the statistical analysis been performed appropriately and rigorously? 

Reviewer #1: Yes

Reviewer #2: Yes

3. Have the authors made all data underlying the findings in their manuscript fully available?

Reviewer #1: No

Reviewer #2: Yes

4. Is the manuscript presented in an intelligible fashion and written in standard English?

Reviewer #1: No

Reviewer #2: Yes

5. Review Comments to the Author

Reviewer #1: The manuscript lacks the novelty component and there are several technical points that makes this study not suitable for publication in the present form. The function of WRINKLED1 gene in context of lipid biosynthesis and their effect on plant growth has been already reported by various researcher previously in different plants. Author only reported the role of rice WRINKLED1 gene lipid biosynthesis and their effect on plant vegetative growth.

My specific comments are as under:

• Information on the confirmation of the integration and homozygosity of transgene in transgenic rice plants is missing.

• Line 189-190:- The author reported that the transcripts level of OsWRI1a in transgenic rice was around 25-35 fold high in leaves tissue (Fig 4D) as compared to control plant leaves. The author generated the OsWRI1a overexpression transgenic rice plant under endosperm specific Bt2 P1 (PBt2P1) promoter (highly expressed in endosperm and lower in leave; line:-174-175). It is not clear as to why OsWRI1a overexpression lines showed elevated OsWRI1a transcript levels in leaves tissue when it is an endosperm specific promoter.

• There is no insight into why OsWRI1a overexpression lines showed retarded plant growth phenotypes.

Reviewer #2: Comments on the manuscript entitled, “Ectopic expression of Wrinkled1 in rice improves lipid biosynthesis but retards plant growth and development”

The authors looked at two rice gene calls with functional annotation with the transcription factor that acts as a global regulator of fatty acid biosynthesis wrinkled1 (Wri1). They demonstrated that as expected the OsWri1a is post translationally targeted to the nuclear compartment using an Arabidopsis GFP fusion platform. The authors then assembled two cassettes to ectopically express OsWri1 constitutively and in a seed preferred fashion. The rationale for doing this is found on line 163 of the manuscript: “Ectopic expression of the WRI1 gene in several plant species results in an increase in oil accumulation in leaves and other organs. To investigate whether ectopic expression of this gene exerts the same effect in rice, we overexpressed the WRI1a gene under the control of the CaMV 35S promoter.”

This is where the problem lies in their communication. There are previous publications relating to expression of the AtWri1 in rice, that include additional alleles to pull carbon towards TAG biosynthesis (AtDGAT/AtPDAT) and protect TAG turnover (AtOle) see Izadi-Darbandi et al 2020 Mol Biol Rep 47:7917, and ectopically express AtWri1 alone in rice see Yang et al 2019 Biosci, Biotechnol, Biochem 83:1807. In the previous publications, significant increase in vegetative tissue fatty acids, with some seed reserve changes, but no phenotypic off types observed. The two key differences between this manuscript and the previous published ones, are ectopic expression of the OsWri1 and the rice genotype. The authors cannot ignore these publications, and will need to address, or at least attempt to explain why the different outcomes. In addition, Mano et al 2019 Plants8, 207; doi:10.3390/plants8070207 communicate on much deeper in silico analysis on rice Wri1 alleles. This publication is also not mentioned in this manuscript.

Secondly, the rationale for targeting lipid biosynthesis in a grain feedstock would be to see if this approach can impact harvest index, and thus grain yield. The outcome of merely stating see what happens, was expected.

6. PLOS authors have the option to publish the peer review history of their article (what does this mean?). If published, this will include your full peer review and any attached files.

Reviewer #1: **Yes: **Arun Kumar

Reviewer #2: **Yes: **Tom Elmo Clemente

---

## [Author Response · Author response to Decision Letter 0]

19 Mar 2022

Dear Editor and Reviewers:

Thank you, Respected Editor and Reviewers, for your valuable comments, and thank you for your precious time dedicated to reviewing this paper. The manuscript has been carefully revised, and all the sections and parts were modified according to the comments. The authors hope that the manuscript in the revised form meets the expectations of the reviewer and can express the content of our research to the community with more clarity. Revised portions of the paper are marked in red. In addition, the paper has been edited by a native-English speaker with a PhD in a relevant discipline working for the professional editing company Sees-editing Ltd. 

We hope you will consider this work for publication in PLos One. Thank you again for your consideration.

Sincerely,

Huawu Jiang

Review Comments to the Author

Reviewer #1: The manuscript lacks the novelty component and there are several technical points that makes this study not suitable for publication in the present form. The function of WRINKLED1 gene in context of lipid biosynthesis and their effect on plant growth has been already reported by various researcher previously in different plants. Author only reported the role of rice WRINKLED1 gene lipid biosynthesis and their effect on plant vegetative growth.

Response: Thank you for your comments concerning our manuscript. Professor, the purpose of this paper is to test whether WRINKLED1 gene can improve the endosperm oil content of starchy crops. The results showed that in the endosperm of starchy crops, increasing the expression of WRINKLED1 could only slightly increase the oil content. At the same time, we further found that increasing the expression of WRINKLED1 affected the growth of leaves of transgenic plants. Finally, thank you very much for your time and effort in reviewing our manuscript.

 My specific comments are as under:

 • Information on the confirmation of the integration and homozygosity of transgene in transgenic rice plants is missing.

Response: Thank you, professor, for your comments and valuable addition. This is an important point that we should have mentioned. Professor, we have added such important information forward to lines 114-118 in the manuscript as you suggested. The contents are as follows: The screening of homozygotes was based on a segregation ratio of about 3:1 observed in T2 plants. The segregation ratios of four transgenic plants, were confirmed by GUS staining, were 237:78, 197:64, 169:55 and 206:67, respectively. Then we randomly selected 30 T3 plants for GUS staining. We speculated that these plants were homozygous plants for all GUS-stained plants.

• Line 189-190: The author reported that the transcripts level of OsWRI1a in transgenic rice was around 25-35 fold high in leaves tissue (Fig 4D) as compared to control plant leaves. The author generated the OsWRI1a overexpression transgenic rice plant under endosperm specific Bt2 P1 (PBt2P1) promoter (highly expressed in endosperm and lower in leave; line:-174-175). It is not clear as to why OsWRI1a overexpression lines showed elevated OsWRI1a transcript levels in leaves tissue when it is an endosperm specific promoter.

Response: Thank you, Professor, for your comments. Although Bt2 gene was mainly expressed in endosperm, it was also weakly expressed in leaves. Therefore, we speculate that using the promoter of Bt2 gene to drive OsWRI1 gene expression will further increase the expression of OsWRI1 gene in the leaves of transgenic plants. In addition, in the endosperm, the blue product produced by GUS reaction appeared after 10 minutes of reaction and reached saturation in 30 min. However, in the leaves, it was difficult to detect GUS activity after 2 hours of reaction, and there was blue product after 4 hours of reaction (Pan X, Jiang H, Yan H, Li M, Wu G. Promoter analysis of the gene encoding ADP-glucose pyrophosphorylase small subunit I in rice. Journal of Tropical and Subtropical Botany. 2008; 16: 189–194 (in Chinese).).

 • There is no insight into why OsWRI1a overexpression lines showed retarded plant growth phenotypes.

Response: Thank you, Professor, for your comments and suggestions. Professor, actually during the revision process of the manuscript we can make changes to the manuscript. With due respect Sir, contribution to the field is an option while submitting the new article. Sir, your suggestion is valuable for us and we will carefully keep this valuable point in our mind for the next time we submissions to PLos One Journals. In addition, because a high level of expression of the OsWRI1a gene (under the control of the CaMV 35S promoter) seriously inhibited rice growth, the transgenic plants died and we couldn't get the seeds. Therefore, we have not carried out the research on the relevant mechanism.

Reviewer #2: Comments on the manuscript entitled, “Ectopic expression of Wrinkled1 in rice improves lipid biosynthesis but retards plant growth and development”

Response: Thank you, Respected Professor, for your valuable comments, and thank you for your precious time dedicated to reviewing this paper. 

• The authors looked at two rice gene calls with functional annotation with the transcription factor that acts as a global regulator of fatty acid biosynthesis wrinkled1 (Wri1). They demonstrated that as expected the OsWri1a is post translationally targeted to the nuclear compartment using an Arabidopsis GFP fusion platform. The authors then assembled two cassettes to ectopically express OsWri1 constitutively and in a seed preferred fashion. The rationale for doing this is found on line 163 of the manuscript: “Ectopic expression of the WRI1 gene in several plant species results in an increase in oil accumulation in leaves and other organs. To investigate whether ectopic expression of this gene exerts the same effect in rice, we overexpressed the WRI1a gene under the control of the CaMV 35S promoter.” This is where the problem lies in their communication. 

Response: Thank you, Professor, for your valuable comments. Professor, we have carefully revised the manuscript according to your suggestions. Thank you very much for your time and effort in reviewing our manuscript.

There are previous publications relating to expression of the AtWri1 in rice, that include additional alleles to pull carbon towards TAG biosynthesis (AtDGAT/AtPDAT) and protect TAG turnover (AtOle) see Izadi-Darbandi et al 2020 Mol Biol Rep 47:7917, and ectopically express AtWri1 alone in rice see Yang et al 2019 Biosci, Biotechnol, Biochem 83:1807. In the previous publications, significant increase in vegetative tissue fatty acids, with some seed reserve changes, but no phenotypic off types observed. The two key differences between this manuscript and the previous published ones, are ectopic expression of the OsWri1 and the rice genotype. The authors cannot ignore these publications, and will need to address, or at least attempt to explain why the different outcomes. In addition, Mano et al 2019 Plants8, 207; doi:10.3390/plants8070207 communicate on much deeper in silico analysis on rice Wri1 alleles. This publication is also not mentioned in this manuscript. Secondly, the rationale for targeting lipid biosynthesis in a grain feedstock would be to see if this approach can impact harvest index, and thus grain yield. The outcome of merely stating see what happens, was expected.

Response: Thank you, Professor, for your valuable comments. Professor, this is an important point that we should have mentioned in the discussion. According to your comments, we have added a discussion of why AtWRI1 and OsWRI1 function differently in rice in the revised MS, in lines 245-251. 

In addition, professor, we have added the references about Mano et al 2019 Plants8, 207; doi:10.3390/plants8070207 in the revised MS. 

Finally, we hope that the manuscript in the revised form meets the expectations of the reviewer and can express the content of our research to the community with more clarity. Thank you again for your consideration.

---

## [Decision Letter · Decision Letter 1]

14 Apr 2022

Ectopic expression of WRINKLED1 in rice improves lipid biosynthesis but retards plant growth and development

PONE-D-22-01261R1

Dear Dr. Jiang,

We’re pleased to inform you that your manuscript has been judged scientifically suitable for publication and will be formally accepted for publication once it meets all outstanding technical requirements.

Kind regards,

Tamar Juven-Gershon, Ph.D.

Academic Editor

PLOS ONE

Reviewers' comments:

Reviewer's Responses to Questions

**Comments to the Author**

1. If the authors have adequately addressed your comments raised in a previous round of review and you feel that this manuscript is now acceptable for publication, you may indicate that here to bypass the “Comments to the Author” section, enter your conflict of interest statement in the “Confidential to Editor” section, and submit your "Accept" recommendation.

Reviewer #1: All comments have been addressed

Reviewer #2: All comments have been addressed

2. Is the manuscript technically sound, and do the data support the conclusions?

Reviewer #1: Yes

Reviewer #2: Yes

3. Has the statistical analysis been performed appropriately and rigorously? 

Reviewer #1: Yes

Reviewer #2: Yes

4. Have the authors made all data underlying the findings in their manuscript fully available?

Reviewer #1: Yes

Reviewer #2: Yes

5. Is the manuscript presented in an intelligible fashion and written in standard English?

Reviewer #1: Yes

Reviewer #2: Yes

6. Review Comments to the Author

Reviewer #1: Authors have addressed my comments satisfactorily. Just found one typo mistake in Figure 3 (Please change Brihgt to Bright).

Reviewer #2: The authors likely observed relatively high miss expression of the Brittle2 promoter given the use of pCAMBIA1301binary vector, for whatever orientation the Brittle2-OsWRI1 cassette was cloned in, the Brittle2 and a 35S CaMV promoter element would reside 5' to 5', which will impact specificity of Brittle2 expression

7. PLOS authors have the option to publish the peer review history of their article (what does this mean?). If published, this will include your full peer review and any attached files.

Reviewer #1: **Yes: **Dr. Arun Kumar

Reviewer #2: **Yes: **Tom Elmo Clemente

---

## [Editor Report · Acceptance letter]

11 Aug 2022

PONE-D-22-01261R1 

Ectopic expression of *WRINKLED1* in rice improves lipid biosynthesis but retards plant growth and development 

Dear Dr. Jiang:

I'm pleased to inform you that your manuscript has been deemed suitable for publication in PLOS ONE. Congratulations! Your manuscript is now with our production department. 

Kind regards, 

on behalf of

Prof. Tamar Juven-Gershon 

Academic Editor

PLOS ONE